

# Emission ensemble approach to improve the development of multi-scale emission inventories

Philippe Thunis[1], Jeroen Kuenen[2], Enrico Pisoni[1], Bertrand Bessagnet[1], Manjola Banja[1]. Lech Gawuc[3], Karol Szymankiewicz[3], Diego Guizardi[1], Monica Crippa[1,4], Susana Lopez-Aparicio[5], Marc Guevara[6], Alexander De Meij[7], Sabine Schindlbacher[8], Alain Clappier[9]

[1] European Commission, Joint Research Centre, Ispra, Italy
[2] TNO, Department of Air, Climate and Sustainability, Utrecht, The Netherlands
[3] Institute of Environmental Protection – National Research Institute (IEP-NRI), Słowicza 32, 02-170 Warsaw, Poland
[4] Unisystem S.A., Milan, Italy
[5] NILU – Norwegian Institute for Air Research, 2027 Kjeller, Norway
[6] Barcelona Supercomputing Center, Barcelona 08034, Spain
[7] MetClim, Varese, 21025, Italy
[8] Environnent Agency Austria, Spittelauer Lände 5, 1090 Vienna, Austria
[9] Université de Strasbourg, Laboratoire Image Ville Environnement, Strasbourg, France

*Correspondence to*: Philippe Thunis (Philippe.THUNIS@ec.europa.eu)

## Abstract

In this work, an ensemble inventory (median) is created with the aim of monitoring the status and progress made with the development of Europe-wide inventories. This ensemble inventory also allows comparing a large number of inventories at the same time, foster interactions among emission inventory developers and allow for comparing additional inventories (e.g. bottom-up ones) with all ensemble components. In contrast with other fields of applications (e.g. air quality forecast), this emission ensemble is not necessarily better than any of its components. Although it is not the more accurate inventory, it serves here as a common benchmark for the screening. We focus on differences in terms of country totals, country sectorial share and share of the country emissions to the urban areas for emissions of $NO_x$, PM2.5, PM coarse, NMVOC, $SO_x$ and $NH_3$. Because the emission "truth" is unknown, the approach does not tell which inventory is the closest to reality. The methodology rather screens differences between inventories, excludes differences that are not relevant and identifies among the remaining ones, those that are larger than a given threshold, and need special attention. The underlying concept is that above this threshold, differences are so large that one or both inventories must be checked.

The analysis of the ensemble and the comparison with its individual components highlight a large number of inconsistencies. While two of the three inventories behave more closely to each other (CAMS-REG and EMEP), they yet show inconsistencies in terms of the spatial distribution of emissions. These differences mostly occur for $SO_2$, PM and NMVOC, for the industrial and residential sectors, and reach a factor 10 in some instances. Necessary improvements have been identified, in particular with EDGAR with the PM emissions from the small-scale combustion sector and $SO_2$ from the industry and power plant sectors. The comparison with the local inventory for Poland leads to identifying another type of inconsistencies, associated to the sectorial share at country level. This is explained by the fact that some emission sources are omitted in the local inventory due to the lacking of appropriate geographically allocated activity data. The screening process led to identify some sectors and pollutants for which discussion between local and EU-wide emission compilers would be needed in order to reduce the



magnitude of the observed differences (e.g. in the residential and industrial sectors). The settings used in
this work (e.g. the choice of 150 urban areas or the way sectors are aggregated) are arbitrarily fixed and
can easily be adapted for the purpose of other comparisons.
**Keywords**: emission inventories, quality assurance, quality control, screening, urban emissions,
ensemble

## 1. Introduction

Ensemble of models have widely been used in climate (Kotlarski et al., 2014) and air quality
modelling fields throughout the world (Stevenson et al., 2006; Vautard et al, 2009; Marecal et al.
2015; Brasseur et al., 2019) providing better and more robust results using a set of model results
instead of relying on a unique realization. While in some instances, reference values (e.g.,
measurements) exist against which models can be compared, this is unfortunately not the case
for emissions, and hence the emission ensemble is not necessarily better than any of its
components. The emission ensemble is therefore not a more accurate inventory but can serve as a
common benchmark to support the assessment of methods to develop spatially resolved emission
inventories.
In Thunis et al. (2022)  we designed a methodology to compare two emission inventories, one
against the other. This methodology was analysing differences the differences between these two
inventories in terms of country totals, country sectorial share and share of the country emissions
to the urban areas (i.e. how much of the country total is allocated to the urban area). In this work
we follow the same principle to analyse differences but we introduce an ensemble concept to
allow comparing a larger number of inventories at the same time.
In addition to this key advantage, several other objectives are pursued by introducing the
ensemble for EU wide emission inventories, namely  (1) to create a unique common benchmark,
based on state-of-art inventories, to monitor and quantify the current level of agreement
associated to these inventories; (2) to identify and characterize the largest mismatches in terms of
pollutant, sector among all ensemble components; (3) to foster interactions between EU wide
emission inventory developers around identified inconsistencies and (4) to allow for comparing
additional inventories (e.g. bottom-up ones) with all ensemble components in a bilateral
approach. Because the emission "truth" is unknown, the approach does not tell which inventory
is the closest to reality. The methodology rather screens differences between inventories,
excludes differences that are not relevant (i.e., large differences on low emission values are
disregarded) and identifies among the remaining ones, those that are larger than a given
threshold, and need special attention. The underlying concept is that above this (arbitrary)
threshold, differences are so large that one or both inventories can be considered wrong. The
choice of this vocabulary, i.e. wrong is intentional and is meant here to foster the process of
reviewing the data when differences exceed a given threshold. In other words, a factor 100
between inventory estimates for a given emission most likely reveals one or more huge errors (or
inconsistencies) that are relatively straightforward to identify and must be addressed in one or
both inventories.
The emission ensemble is also intended as a focal point for inter-comparisons against which
bilateral analyses can take place (one inventory against the ensemble), with the aim to improve





the benchmark and assessment. The main advantage is to structure the inter-comparison process
around a single benchmark, in our case the ensemble, rather than by organizing a series of
disconnected inter-comparisons (inventory 1 vs. inventory 2, inventory 2 vs inventory 3…).
Finally, it  supports discussions among emission compiling teams on the main inconsistencies,
methodologies behind compilations, and gain understanding about the main reasons for
differences, with a view to resolve them and progressively improve emission inventories.
When inconsistencies are identified among EU wide inventories, a comparison of the ensemble
with local (intended here as national or sub-national) inventories can be helpful, as local scale
information is an independent source of information, which methods are based on local
knowledge and understanding of the activities that result on emissions.
The work is structured as follows. In Section 2, we review the screening methodology proposed
in Thunis et al. (2022) and discuss the problematic of introducing an ensemble in the frame of
this screening approach. In Sections 3, we apply the screening approach to the European-wide
inventory components of the ensemble whereas we illustrate in Section 4 how this ensemble can
then be compared to local inventories in a bilateral manner. For the latter, the Poland local
inventory is used. In Section 5, we discuss the main findings from both type of comparisons and
conclude in Section 6.

## 2. Description of the methodology

### 2.1   Overview of the screening methodology

In this section, we provide a brief summary of the screening method detailed in Thunis et al.
(2022). The approach aims at comparing two emission inventories over a series of urban areas
over which the consistency is assessed for all sectors and pollutants. Based on gridded yearly
emission inventories detailed in terms of emitted pollutants (denoted as "$p$") and sectors of
activity (denoted as "$s$"), the data required for each pollutant and sector (denoted as a $[p,s]$
couple) are twofold and consist of (1) emissions aggregated over specific urban areas (denoted
by a lowercase notation $e_{p,s}$) and country scale emissions (denoted by an uppercase notation
$E_{p,s}$).
Consistency is assessed around three aspects: (1) the total pollutant emissions assigned at
country level; (2) the way these country emissions are shared in terms of sector of activity and 3)
the way country scale emissions are distributed to the urban areas. To address these three
aspects, we decompose the ratio of the known pollutant-sector emissions for each city as follows:

$$\frac{e_{p,s}^1}{e_{p,s}^2} = \frac{\dfrac{e_{p,s}^1}{E_{p,s}^1}}{\dfrac{e_{p,s}^2}{E_{p,s}^2}} * \frac{\dfrac{E_{p,s}^1}{\bar{E}_p^1}}{\dfrac{E_{p,s}^2}{\bar{E}_p^2}} * \frac{\bar{E}_p^1}{\bar{E}_p^2} \qquad (1)$$






where $\bar{E}_p$ represents the country scale emissions summed over all sector for a given pollutant.
Superscripts refer to the two inventories used for the screening. Equation (1) is an identity where
all terms are known from input quantities, i.e. the city and country scale emissions detailed in
terms of pollutants and sectors. The three terms on the right-hand side of the identity provide
information on the urban share (denoted as *FAS* for Focus Area Share*)*, on the country sectorial
share (denoted as *LSS* for Large Scale Sectorial share*)* and on the country pollutant totals
(denoted as *LPT* for Large scale pollutant Total*)*.
For convenience, we rewrite equation (1) in logarithm form as:

$$log\left(\frac{e_{p,s}^1}{e_{p,s}^2}\right) = log\left(\frac{\frac{e_{p,s}^1}{E_{p,s}^1}}{\frac{e_{p,s}^2}{E_{p,s}^2}}\right) + log\left(\frac{\frac{E_{p,s}^1}{\bar{E}_p^1}}{\frac{E_{p,s}^2}{\bar{E}_p^2}}\right) + log\left(\frac{\bar{E}_p^1}{\bar{E}_p^2}\right) \qquad (2)$$

Which can be rewritten as equation (3) with simplified notations:

$$\hat{e} = \widehat{FAS} + \widehat{LSS} + \widehat{LPT} \qquad (3)$$

where the hat symbol indicates that quantities are expressed as logarithmic ratios. These three
quantities are at the basis of the screening methodology and serve as input for the graphical
representation as well.
Because the number of [*p,s*] points under screening, equal to the product of the number of
pollutants by the number of sectors itself multiplied by the number of urban areas (i.e. $N \times N_p \times$
$N_s$), may become overwhelming, we proceed with a number of steps that help focusing the
screening on priority aspects. First, we restrict the screening to emissions that are relevant, i.e.
large enough (in practice the condition $e_{p,s}/\overline{E_p} > \gamma_t \times \max_{p,s}\{e_{p,s}/\overline{E_p}\}$ is tested for each (p,s)
couple with a user threshold parameter set by the user, $\gamma_t$). As shown in Thunis et al. (2022), this
exclusion step with $\gamma_t$ =0.5 leads to eliminating a large fraction of the [*p,s*] couples from the
screening process (between 80 and 90%). Second, we flag, among the remaining relevant
emissions, only those for which inventory differences in emissions are larger than a given
threshold ($\beta_t$).
Differences originate from methodological choices but also from errors generated during the
inventory compilation process. When differences are small, it is not possible to tell whether they
originate from methodological choices or from errors. We refer to these small differences as
"uncertainty". Although very large differences may result from methodological choices as well
(e.g., inclusion or not of particulate matter condensable emissions for the residential sector), they
are more likely to be associated to errors. Given the magnitude of the differences, it will in most
cases be possible to identify one best value out of the two inventory estimates, even though the
true emissions are unknown. These large differences are named "inconsistencies". In the
proposed screening methodology, a threshold of 2 (free parameter) is introduced to distinguish
inconsistencies from uncertainties.



As a follow-up step, all [p,s] couples that remain after the relevance ($\gamma_{p,s} > \gamma_t$) and
inconsistency detection steps ($\beta_{p,s} > \beta_t$), are used to calculate an "Emission Consistency
Indicator (ECI)" as follows:

$$ECI = \max_{\{relevant\ emissions\}} \frac{\log(\beta_{p,s})}{\log(\beta_t)} \tag{4}$$

The ECI quantifies the maximum difference among all relevant [p,s], normalized by the
inconsistency level ($\beta_t$). It therefore quantifies the ratio between the maximum inconsistency and
the assumed level of uncertainty. A value of ECI less than one means that all differences are
considered as uncertainty (in other words none of the inventory can be identified as best
performing). Together with the ECI, which quantifies this maximum difference, we associate the
percentage of inconsistent [p,s] with respect to the total number of relevant data, to provide
information on the number of detected inconsistencies. To facilitate the screening process, these
concepts are displayed graphically.
Finally, we prioritise inconsistencies following the LPT – LSS – FAS hierarchy. In other words,
if large scale inconsistencies are spotted for LPT, they are flagged as the priority, regardless of
the magnitude of inconsistencies calculated for LSS and/or FAS. If no inconsistency is flagged
for LPT, the same holds for LSS regardless of the level of inconsistency calculated for FAS.
Consequently, the inconsistency flagged as priority might not be the largest inconsistency. This
hierarchy is motivated by the fact that addressing large scale inconsistencies will lead to
potentially resolving many issues at small scale at once (all urban areas within a given country).
Inconsistencies are counted when the individual terms in equation (3) are larger than the
threshold $\beta_t$ but also when the indicators sums (i.e., $\widehat{FAS} + \widehat{LSS} + \widehat{LPT}, \widehat{LSS} + \widehat{LPT}$) exceed this
threshold.
It is important to note that the approach follows a bottom-up approach, i.e., we assess the three
types of inconsistencies for each city, pollutant and sector. This means that the same LPT
inconsistency are counted for all cities within a given country or for all sectors for a given
pollutant. Similarly, a LSS inconsistency is counted for each city belonging to the same country.
While this might be seen as double counting of some inconsistencies, the approach allows
comparing local vs country scale indicators.
## 2.2   Construction of an ensemble as reference
This work aims at applying the ensemble concept to extend the Thunis et al. (2022) methodology
to several inventories. The ensemble is calculated from EU-wide inventories that have been
developed and regularly updated over several years within the EU[1]. While either the mean or the
median of these inventories could be used to calculate the ensemble, we here use the median as it
has been shown to be a more robust indicator than the mean (Riccio et al. 2007). Indeed, if one
of the inventories is a strong outlier (i.e., much larger or much smaller values), the mean would
be strongly influenced by these extreme values and would differ from the values of the majority

---

[1] Note that EDGAR is designed as a global inventory but we consider here its European coverage only in this analysis and refer to it as a European wide inventory





of the inventories. On the other hand, the median is not affected by extreme values and therefore
takes a value closer to the values taken by the majority of the inventories. It therefore remains
further away from outliers, which become easier to identify.
In this work, the ensemble is created from three state-of-the-art Europe wide inventories: CAMS-
REG, EMEP and EDGAR (see details in following section) and is defined on a yearly basis by
taking values of the year of interest. Urban ($e_{p,s}$) and country emissions ($E_{p,s}$) for the selected
year are required as input. Independent ensemble values are defined for each $E_{p,s}$ and $e_{p,s}$ as the
median of the three inventory values. For a given area, the urban and country scale emission
ensembles for a given year read as:

$$
\begin{aligned}
e_{p,s}^{ens} &= median\left\{e_{p,s}^{CAMS}, e_{p,s}^{EMEP}, e_{p,s}^{EDGAR}\right\} \\
E_{p,s}^{ens} &= median\left\{E_{p,s}^{CAMS}, E_{p,s}^{EMEP}, E_{p,s}^{EDGAR}\right\}
\end{aligned}
\tag{5}
$$

Note that this calculation implies that $e_{p,s}^{ens}$ and $E_{p,s}^{ens}$ might not belong to the same inventory for a
given area and pollutant-sector couple [p,s]. It is also worth mentioning that should one
inventory behave as an outlier, its value will not be selected in the ensemble.
The proposed approach then consists in comparing each inventory with the ensemble to identify
inconsistencies. This generalization of Thunis et al. (2022) leads to the same kind of conclusions
where inconsistencies most likely highlight errors in the flagged inventory, but it is however not
possible to exclude that the inconsistency originates from the ensemble (i.e., be present in all
other inventories). Despite this inconveniency, the method remains an efficient way to identify,
among the large amount of data from several inventories, those that are most likely to be
problematic and therefore need to be verified in priority.

## 3. Application to EU-wide inventories
### 3.1  Input data
The screening methodology is applied to three state of the art inventories: CAMS-REG v5.1,
EDGAR v.6.1 (Crippa et al. 2022) and EMEP (2022 gridding) that cover emissions for Europe
for the main air pollutants. Urban areas are defined as functional urban areas (FUA, OECD
2012) for which emissions ($e_{p,s}$) are obtained by aggregating grid cell values over these areas.
The FUA is composed of a core city plus its wider commuting zone, consisting of the
surrounding travel-to-work areas. About 150 FUAs across Europe are selected for this screening.
Details on these cities are provided in Thunis et al. (2018). The larger scale emissions ($E_{p,s}$) are
defined at country level, level at which emissions are initially reported for these emission
inventories.
In terms of pollutants, $E_{p,s}$ and $e_{p,s}$ include the following: NO$_x$, NMVOC, PM$_{2.5}$, PM$_{co}$ (coarse
PM, calculated as the difference between PM$_{10}$ and PM$_{2.5}$ emissions), SO$_2$ and NH$_3$, whereas
sectors are based on the Gridded Nomenclature For Reporting (GNFR) classification (NFR-I,
2023 and Table 1 in supplementary material). The original GNFR sectors have been aggregated



in 5 categories: road transport (F), residential (C), power plants (A), industry (B) and others. The
latter category includes fugitive emissions (D), solvents (E), shipping (G), aviation (H). off-road
transport (I), waste (J) and agriculture (K-L). The reference year for all three inventories is 2018.
Finally, the threshold to distinguish relevant from non-relevant emissions as well as the threshold
to distinguish uncertainties from inconsistencies are set to 0.5 and 2, i.e., $\gamma_t$=0.5 and $\beta_t$=2.

CAMS-REG version 5.1 is an emission inventory developed as part of the Copernicus
Atmosphere Monitoring Service (CAMS) to support European scale air quality modelling
(Kuenen et al. 2022). The inventory builds on the officially reported emission data to EMEP in
the year 2020, which are complemented by other sources where reported data are not available or
deemed of insufficient quality. The data are spatially distributed consistently across the entire
domain at a resolution of 0.05x0.1 degrees (lat-lon). The spatial distribution takes into account
specific point source emissions as reported in the European Pollutant Release and Transfer
Register (EPTR2022) to correctly represent point source emissions to the extent possible. The
emissions are provided in GNFR format. The emission dataset is used in support of the CAMS
regional modelling activities, but is also publicly available to support air quality assessment at
European level. CAMS-REG-v5.1 is an update of version 4.2 (which is extensively described in
Kuenen et al. 2022), the main difference being the latest version based on the official
submissions of national emission inventories in the year 2020.

EDGAR is a global emission inventory providing country and sector specific greenhouse gas and
air pollutant emissions from 1970 till nowadays. EDGAR is becoming a global reference in the
field of anthropogenic emissions, in particular contributing to the IPCC AR6 and to the yearly
UNEP emissions gap report (UNEP2021) tackling global climate change issues. In the context of
air pollution, EDGAR is also widely used by air quality modellers and in particular is used as
gap-filling inventory in the context of the Hemispheric Transport of Air Pollution mosaic
compilation. Emissions are computed using a consistent methodology for all world countries,
following the IPCC Guidelines (IPCC 2006, 2019) and EMEP/EEA Guidebook (EMEP/EEA,
2016, 2019) for greenhouse gases (GHGs) and air pollutants, respectively. Emissions are
computed for all IPCC anthropogenic emitting sectors, with the exception of Land Use, Land
Use Change and Forestry, making use of international statistics and default emission factors
complemented with state-of-the-art information. Annual sector and country specific emissions
are then downscaled over the globe at 0.1x0.1 degree resolution making use of hundreds of
spatial proxies. Details about the EDGAR methodology and the assumptions behind the spatial
data used to downscale national emissions are available in several scientific publications
(Janssens-Maenhout et al. 2015, 2019; Crippa et al. 2018, 2021; Crippa et al. 2020; Oreggioni et
al. 2022). Annual emission data are further disaggregated into monthly emissions to further
support atmospheric modellers in simulating the seasonality of anthropogenic emissions (Crippa
et al. 2020).

The EMEP-GNFR (Gridded Nomenclature For Reporting) emissions (Mareckova et al., 2017),
based on 2017 reporting, are compiled within the "UNECE co-operative programme for
monitoring and evaluation of the long-range transmission of air pollutants in Europe", or also
known as EMEP. EMEP is a scientifically based and policy driven programme under the
Convention on Long-range Transboundary Air Pollution (CLRTAP) for international co-
operation, that has the final aim of solving transboundary air pollution problems. Emissions are





built from officially reported data provided to CEIP (Centre of Emission Inventory and
Projection by the Member States in Europe) and follow the EMEP/EEA guidebook guidelines
(EMEP/EEA 2019) to define the annual totals. The emissions are gap-filled with gridded TNO
data from Copernicus Atmospheric Monitoring Service (CAMS) and EDGAR. The dataset
consists of gridded emissions for $SO_x$, $NO_x$, NMVOC, $NH_3$, CO, $PM_{2.5}$, $PM_{10}$ and PMcoarse at
0.1° x 0.1° resolution. More information on the emissions and where to download can be found
in the User Guide (https://emep-ctm.readthedocs.io/en/latest/) and in Mareckova et al., (2017).
The EMEP domain covers the geographic area between 30°N-82°N latitude and 30°W-90°E
longitude.
As these three emission inventories are characterised by different grid resolutions and sector
aggregations, harmonisation is required prior to the screening process for a meaningful
comparison. This has been done in 2 steps:
- by grouping the initial emission categories to common categories, based on GNFR
sectors;
- by aggregating gridded emissions on common polygons, representing cities and
countries.
After this process, emissions inventories can be easily compared among each other.
## 3.2   Results
The first objective of the emission ensemble is to monitor and quantify the current level of
uncertainties/inconsistencies associated to EU-wide inventories, and identify where large
differences come from, in terms of pollutant, sector and location. To perform this task, we apply
the screening methodology by comparing bilaterally each of the three inventories to the
ensemble and report the results in Figure 1 (top left). In this figure, only inconsistencies are
shown, i.e., for emissions that are relevant (i.e., large enough values) for which differences
between inventories are larger than a factor 2 ($\beta_t = 2$). Symbols are used to differentiate
inventories while colours are used to distinguish sectors.
The summary report (bottom part of the top-left figure) provides overview information about
inconsistencies. More than 21% of the relevant emissions ratios show inconsistencies. The ECI
indicator is equal to 132, meaning that the largest inconsistency is more than two orders of
magnitude larger than the level associated to uncertainties. In our case, the EDGAR inventory is
flagged for two thirds of them (227 out of 357), with the largest part of them associated to
industry for $SO_2$ and $PM_{co}$. Inconsistencies are mostly originating from the urban allocation
process (218) but an important number of them also originates at country scale (80+59). It is
important to remember that flagging one particular inventory does not necessarily indicates that
this inventory is the problematic one. But this flagging means that this inventory and/or the
others show an important inconsistency for that city, pollutant and sector which requires further
checking.
In addition to providing a useful summary that details the current state of variability, the diagram
can also serve as basis to monitor progress, through the ECI indicator and associated percentage.



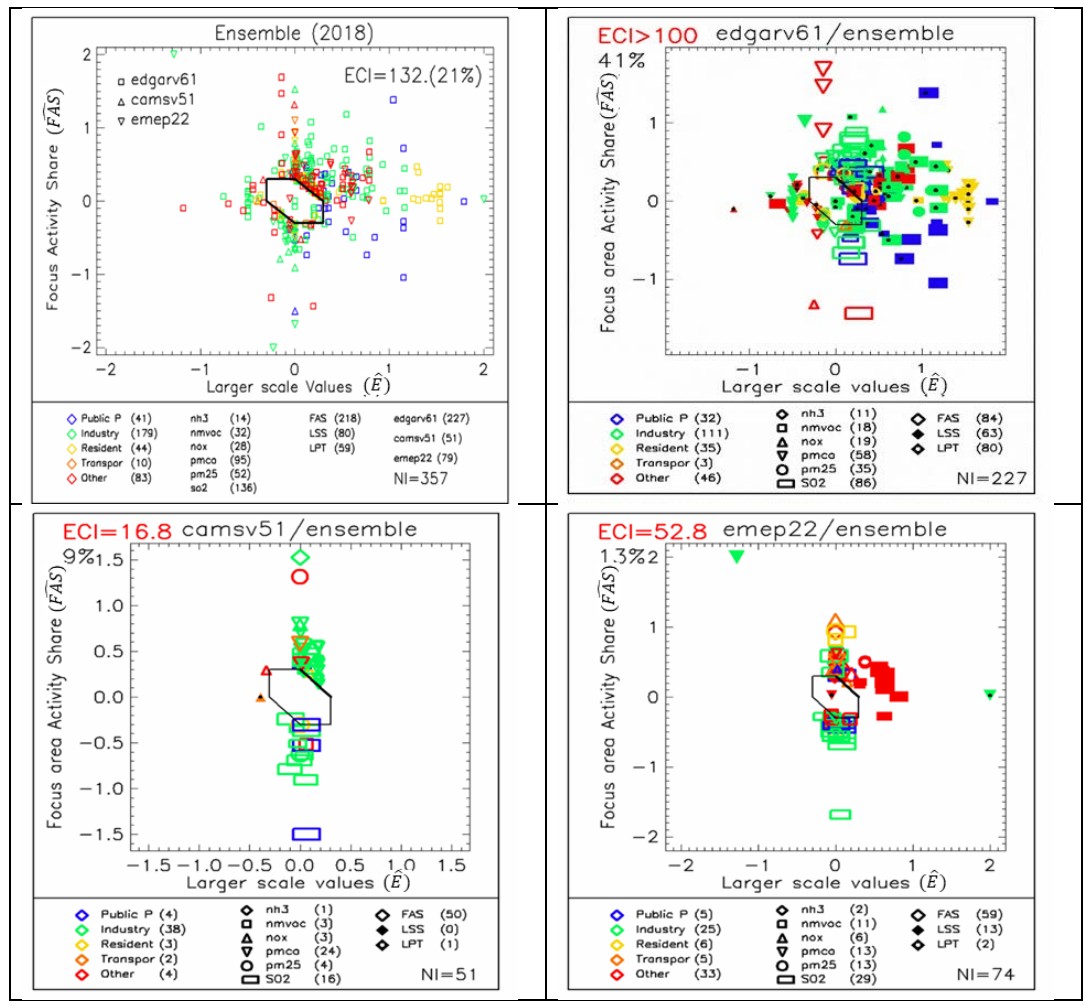

Figure 1: Overview diamonds. The top-left diagram shows the comparison of the three ensemble components (CAMS-REG, EDGAR, EMEP) with the ensemble for 2018. The three following pictures isolate the bilateral comparison of each ensemble component with the ensemble. Symbols and colours are as specified in the legend. Please note that these symbols/colors differ for the top-left panel, compared to the three others. In all diagrams, only inconsistencies are displayed. For visualization purposes, we limit the axis to a factor 2 in terms of magnitude (from -2 to 2) and bound the ECI to 100 (e.g. values of ECI larger than 100 are plotted with a value of 2)

A bilateral comparison of each inventory against the ensemble provides additional information.

For EDGAR, the ECI (>100) indicates that the maximum inconsistency is at least a factor 100 larger than the estimated level of uncertainty (a factor 2 in our case, a value below which differences are assumed to result from uncertainties and small errors, see Section 2.1). Moreover, about 41% of the relevant emission points (large enough emissions) show an inconsistency (difference larger than a factor 2). As indicated by the overview table, these 41% amount to 227 inconsistencies that are shared into about 35% (84) originating from the urban share and 65% originating from country scale issues (83+80), mostly for $SO_2$, $PM_{co}$ and $PM_{2.5}$ from the industry



sector. There are also an important number of inconsistencies related to the "other" (46),
residential (35) and public power sectors (32). In general, for all inconsistencies, EDGAR
estimates are larger than the ensemble ones (all points on the right and/or top of the diagram).
In Figure 2 we identify the most important inconsistency for each city (left side) as well as the
largest inconsistencies (right side) for each of the three right-hand-side terms in equation (3), i.e.
LPT (country pollutant total), LSS (country sectorial share) and FAS (spatialisation).

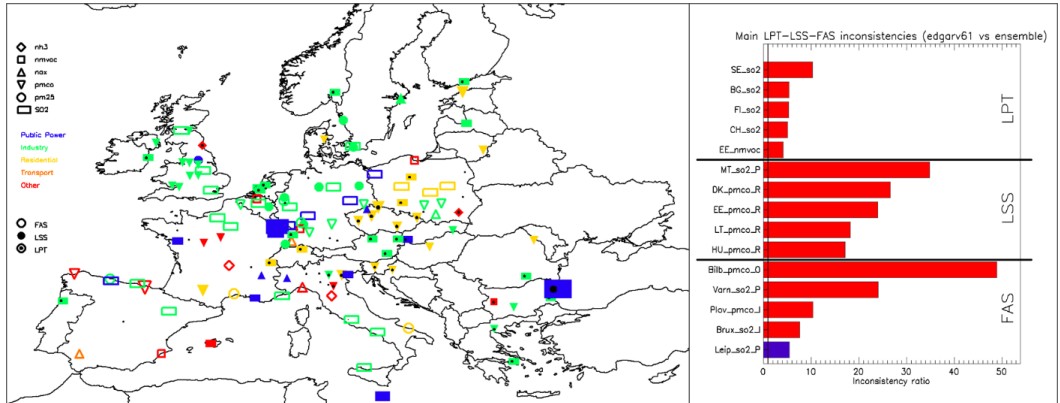

*Figure 2: Left: Main inconsistencies spotted at urban scale for EDGAR when compared to the ensemble (2018). Only the main*
*inconsistency for each city is plotted. See explanation of symbols on the top left of the figure. Right: Major LPT (top 5), LSS*
*(middle 5) and FAS (lower 5) inconsistencies. The two first letters indicate the country code for LSS and LPT whereas the 4 first*
*city letters are given for FAS. Red shading indicates an overestimation and blue shading an underestimation for the EDGAR*
*inventory.*
These figures point to the following main issues:
•    Inconsistencies in $SO_2$ country totals (LPT) in Sweden (factor 10), Bulgaria, Finland and
Switzerland (factor 5). In the case of Sweden and Finland the main difference comes from
the industry sector and especially from the pulp, paper and print sub-sector, for which the
inclusion of black liquor use for energy purposes in EDGAR is the main factor for
differences[2]. EDGAR activity data related to the black liquor statistics need to be revised.
For Bulgaria, the $SO_2$ total is dominated by the public power sector for which the activity
data, sourced from IEA energy balances is subject to regular updates, influencing the
magnitude of the differences. According to IIR 2022 for Bulgaria, SO2 emissions are
regularly updated with measurements, which is not the case of the EDGAR estimations,
explaining part of the differences. Work is in progress to update $SO_2$ abatement measures in
EDGAR. Another issue relates to the application of different emission factors for $SO_2$ that
are based on the sulphur content of fuels, usually not reported regularly by countries, values

---

[2] In Sweden (IIR 2022), the use of black liquor is not applied for energy purposes, whereas in
Finland IIR 2022 a revised methodology for the estimation of SOx-NOx emissions has been
performed which resulted in lower country-specific emission factors.





which are used in CAMS-REG and EMEP[3]. In EDGAR the $SO_2$ emission factors for power
sector has been revised taking into account the limits established by the implementation of
the large Combustion Directive (Directive 2001/80/EC). Slightly different is the situation in
the industry sector where $SO_2$ emission factors for some fuels need to be revised.

•  A larger sectorial share (LSS) at the country level for $SO_2$ in Malta for Public Power (factor
30), for residential PMco emissions in Denmark, Estonia (above a factor 20) and Lithuania
and Hungary (about a factor 10). The large differences in the residential sector between
EDGAR and the other inventories based on country reported values is linked to the estimate
of biomass, both in terms of technology allocation and emission factors applied.  The
EDGAR estimates need to be updated, especially in terms of technology allocation. Although
the filter on low emission values is applied, it is not effective in the case of Malta because it
is a small country where national totals are composed of few power plants only.  The large
LSS ratios obtained there are not significant as the values estimated for the power plant
sector appear to be very small.

•  A few large inconsistencies also appear at the local scale (FAS) due to the use of different
proxies to spatially distribute emissions. This is the case for PMco for the "other" sector in
Bilbao (factor 50). This can probably be explained by the approach followed for the waste
sector for which all emissions are distributed over a few locations only,  using E-PRTR
locations for landfilling and incineration and population in case of missing information. This
results in large differences among inventories due to the proportion of the emissions being
placed within the city area (see Figure 1 in supplementary material). A similar issue appear in
Varna for $SO_2$ for public power (factor >20). Work is in progress to update the spatial
allocation of the public power and waste sectors emissions.

For CAMS-REG, the ECI (=16.8) indicates that the largest inconsistency is around a factor 15
larger than the estimated level of uncertainty. About 9% of the relevant emission points show an
inconsistency larger than a factor 2. As indicated by the overview table, these 9% amount to 51
inconsistencies that are almost all related to urban share issues (50), mostly for $PM_{co}$ and $SO_2$
from the industry sector.

---

[3] The default EMEP/EEA Guidebook 2019 emission factor for $SO_2$ are w/o abatements and only
for 1% mass sulphur content for coal and oil and 0.01 g/m3 for gas (EMEP/EEA guidebook
2019).





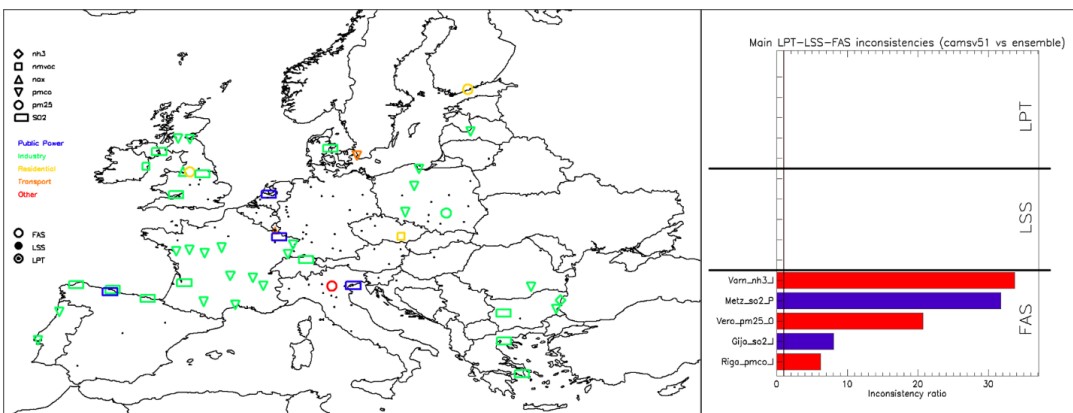

*Figure 3: same as Figure 2 but for CAMS-REG*

Figure 3 points to the following main issues:

- All major inconsistencies are related to the choice of spatial proxies (FAS) and occur in particular in Varna for industrial NH$_3$, in Metz for SO$_2$ from power plant and in Verona for PM2.5 from the "other" sector. These three inconsistencies exceed a factor 20. Note also that these inconsistencies are either over- or under-estimations (red and blue color bars, respectively). In Bulgaria, the largest industrial point source in E-PRTR (68% of the country total) is located near Varna, hence the high emissions there. The large differences among inventories occur due to the proportion of these emissions being placed within the city area (see Figure 2 in supplementary material). The same explains the differences for SO$_2$ in Metz for the power plant sector or for PM2.5 in Verona for the "other sector".

- Although of lower importance, inconsistencies are also spotted for industrial PMco emissions in France and are systematic in several cities across the country. The same occur for industrial SO$_2$ emissions in the UK and in Spain. The diamond plot shows that while PMco has larger estimates in the CAMS-REG inventory, the opposite is true for SO$_2$. A likely explanation for the differences in SO$_2$ emissions is that their attribution to point sources is done only for those included in point source reporting (E-PRTR). Smaller sources which are below the threshold for E-PRTR reporting are distributed as diffuse sources to industrial zones (land cover class). This may lead to over-allocation in some urban areas.

For EMEP, the ECI (52.8) indicates that the maximum inconsistency is about a factor 50 larger than the estimated level of uncertainty. About 13% of the relevant emission points show an inconsistency. As indicated by the overview table, these 13% amount to 74 inconsistencies that are mostly related to the spatial share of the emissions (FAS=59), mostly for SO$_2$ (29), and in a lesser extent to PM$_{2.5}$ (13), PM$_{co}$ (13) and NMVOC (11) originating from the "other" sector (33), but also from the industry (25) sectors.



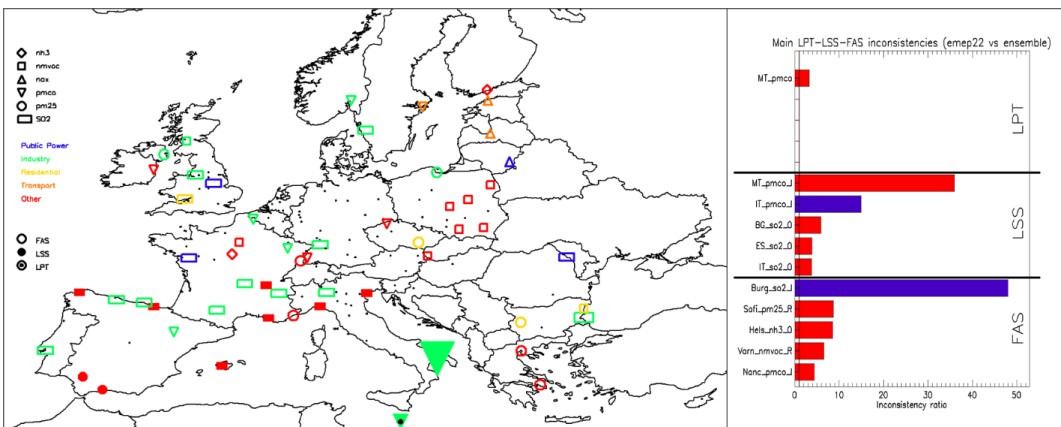

*Figure 4: same as Figure 2 but for EMEP*
Figure 4 points to the following main issues:

• One inconsistency only is spotted at country total level (LPT) for the PMco industrial
emissions in Malta (factor 3). Similarly to what reported for EDGAR for Malta, the low-
emission filter is not efficient to remove these small (not relevant) emissions, given the small
size of the country
• A series of inconsistencies are associated with the sectorial share at country level (LSS). The
largest is observed for PMco industrial emissions in Malta (factor >30) and add up to the
inconsistency at country total level previously highlighted. The same inconsistency, although
as underestimation (blue shaded bar in figure 4), occurs in Italy with a factor 15. LSS
inconsistencies also occur for $SO_2$ emissions from the "other" sector in countries like
Bulgaria, Spain and Italy (between a factor 3 and 6)
• Regarding inconsistences related to spatial proxies, one large one (factor >50) is flagged in
Burgas for $SO_2$ emissions from the industry sector (see Figure 3 in supplementary material).
This type of inconsistencies also occur in a lesser measure in other cities and similarly to
CAMS-REG, are likely explained by the precision of their attribution as point sources.
## 4. Application to local inventories
### 4.1   Input data
In this section, we use the local inventory for Poland and compare it to the Europe wide
ensemble.

The Central Emission Database (CED) is a local emission inventory designed for Polish national
air quality modelling. The CED is based on source location and provides accurate resolution-free
data, which can be gridded depending on the requested target resolution for different
computational grid configurations over Poland (typically 2.5 km over the entire country and 0.5
km for agglomeration zones). The majority of data is processed with respect to its exact
geographical localisation. The intention behind CED is to include documented emission sources
in Poland. Since the inventory is fairly new (the first version was ready in 2019), priority was



given to the most critical sectors, like residential combustion (described in detail in Gawuc et al.,
2021) and road transport. The road transport data presented in this paper (topical for 2019) was
based on traffic models for the major roads in the country. Emissions on minor roads were
distributed using the residue values taken from subtracting emission on major roads from the
national totals. Current methodology (topical for 2022) is based on smartphone car navigation
app which provides GPS data on road traffic and annual average car speed.

One of the essential components of CED is the "National database on greenhouse gases and
other substances emission" (so-called national database – NB). NB consists of information on
installations and sources' location responsible for emission into the atmosphere. NB has
similarities to E-PRTR, but unlike it, it covers all emission sources regardless of type, power or
production level. Registered NB users provide information on emission volumes resulting
directly from the exploitation of their installations, as well as ancillary processes, which may
cause fugitive emissions. NB users may rely on direct stack measurements (continuous or
periodic) in case of more significant emitters. To be applied for CED and air quality modelling,
the reported data is categorized into SNAP and converted to GNFR if needed (Table 1,
supplementary material).
NB is a basis for GNRF A, B, D, E, and J emission estimations contributing to CED. Two
approaches are applied to evaluating CED data. Firstly, as part of each modelling stream (i.e.,
operational air quality forecast, annual air quality assessment, station representativeness
analysis), a comprehensive evaluation is undertaken (station-by-station time series for over 100
monitoring sites for each pollutant). Moreover, spatial patterns of the increments calculated in
the assimilation procedure let to identify and improve the assumptions behind CED. The
database is updated every year and there is a continuous attempt to improve emission estimates
both – for total load and spatial distribution of sources. Modelling results helped to identify
missing sources (e.g. resuspension, underestimated agriculture sector, domestic water heating).
All sectors in CED are constantly improved using the best available activity data.

The comparison between CED and ensemble data is performed on 14 cities, 5 sectors and 6
pollutants, leading to 420 emission ratios being tested. Among these 420 available data, 84 only
remain after the relevance test ($\gamma_t > 0.5$). These 84 [p,s] points serve as basis to identify
inconsistencies ($\beta_t > 2$).

Note that although the year of comparison differs (2018 for the ensemble vs. 2019 for the Polish
emission data), inconsistencies are generally large enough to justify explanations other than
those originating from the difference in terms of reference year.
We first assess how well the Europe wide emission ensemble components agree over Poland and
identify the main inconsistencies from a EU-wide perspective. In a second step, we use local
information to (1) help solving the inconsistencies identified at European level and (2) identify
additional inconsistencies between the ensemble and the local inventory.

## 4.2  Results
Figure 5 displays a zoom of Figure 1 over Poland, focusing on Europe-wide inventories only.
Inconsistencies (Figure 5 top left) occur for about 13% of the relevant [p,s] points, with a
maximum inconsistency (ECI) 2.5 times larger than the assumed level of uncertainty. As seen



from the overview table, most of the issues are related to the EDGAR (20) and EMEP (6)
inventories, in particular to the "residential" sector for EDGAR (Figure 5 top right), to the
industry sector for CAMS-REG (Figure 5 bottom left) and to the "other" sector for EMEP
(Figure 5 bottom right).

0



Figure 5: Overview diamonds. The top-left diagram shows the comparison of the three ensemble components (CAMS-REG, EDGAR, EMEP) with the ensemble inventory over Poland. The three following pictures isolate the bilateral comparison of each ensemble component with the Ensemble. Symbols and colours are as specified in the legend. Please note that these symbols/colors differ for the top-left panel, compared to the three others. In all diagrams, only inconsistencies are displayed.





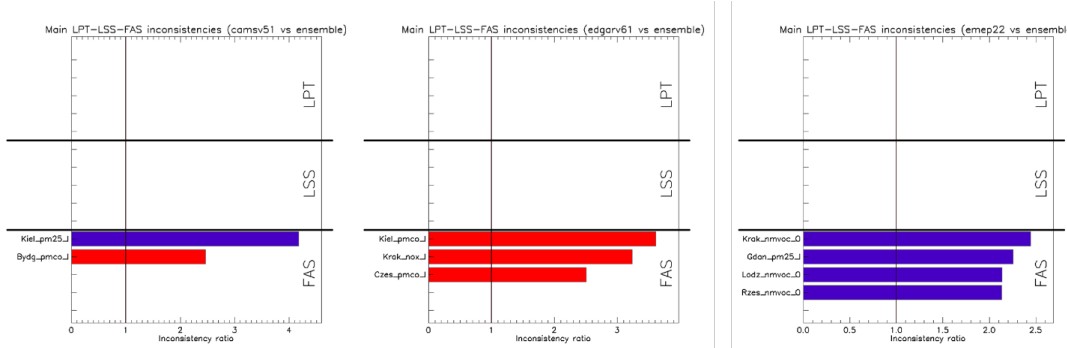

*Figure 6: Major inconsistencies (up to 5 per category) for LPT, LSS and FAS for CAMS-REG (left), EDGAR (middle) and EMEP*
*(right). Red and blue shadings indicate an overestimation or underestimation of the individual inventory with respect to the*
*ensemble, respectively.*
For EDGAR, while Figure 5 indicates a comparable share between country and urban scale
inconsistencies, these country inconsistencies appear because the sum of LPT and LSS is larger
than the threshold of 2 while their individual values remain below this threshold. This is why no
country scale issues appear in Figure 6. The largest (factor 3) urban scale issues (FAS) are
identified for the industrial sector for PMco in Kielce and Czestochowa and for NOx in Krakow.
Gridded data for PMco/Kielce are shown in Figure 4 of the supplementary material. While the
industrial locations are quite similar with those of EMEP and CAMS-REG, EDGAR emission
estimates are much larger in the case of EDGAR.
For EMEP, inconsistencies are all related to the urban share of the emissions (FAS) with factors
slightly larger than 2 for the "other" sector NMVOC emissions in the cities of Krakow (Figure 5
supplementary material), Lodz and Rzeszow and for the PM2.5 industry emissions in Gdansk.
Here again, the localization of the main emission sources is similar with EDGAR and CAMS-
REG, the EMEP estimates are significantly lower.
Similar to EMEP, all inconsistencies in CAMS-REG are related with the spatial share of
emissions. The largest inconsistencies occur for industrial emissions of PM2.5 in Kielce (Figure
6 supplementary material) and of PMco in Bydgoszcz. In both cases, CAMS-REG distributes its
emissions over more locations with a higher intensity.
EDGAR also shows different values in the residential sector for PM2.5 at country level.
Explanations for such differences are linked with the fact that no emissions are allocated to
biomass technologies in EDGAR, and that emission factors for some fuels are very different. For
example, the EDGAR emission factor for other bituminous fuel allocated to small boilers is
nearly the double of the default values. On the other hand, the values reported for Poland (2020)
for both coal and biomass emission factors are well below default values, increasing the
difference with the EDGAR estimation. Note that these emission factors have been significantly
revised in the Poland 2022 submission, which will be reflected in future EMEP and CAMS-REG
inventories.
Next, we check whether the local inventory flags similar and/or other issues. The diamond
diagram in Figure 7 displays a comparison of the local (CED) and Europe wide ensemble




(CAMS-REG, EDGAR, EMEP) inventories for all relevant sector-pollutant points for all cities
in Poland. Out of the 420 emission ratios being tested, only 73 are associated to relevant
emissions among which 49 (i.e. 67%) are identified as inconsistencies. The consistency indicator
(ECI) is around 14, indicating that the maximum inconsistency is larger than the assumed level
of uncertainty by a factor 14. The summary table (at bottom of the diamond) points to the
residential and "other" sectors as the main issues with NMVOC and PM$_{2.5}$ in terms of pollutants.
Most inconsistencies originate at country level, in majority in terms of sectorial share.

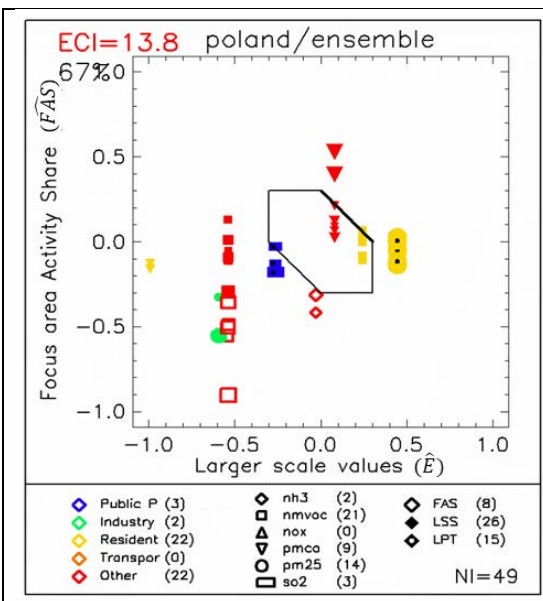
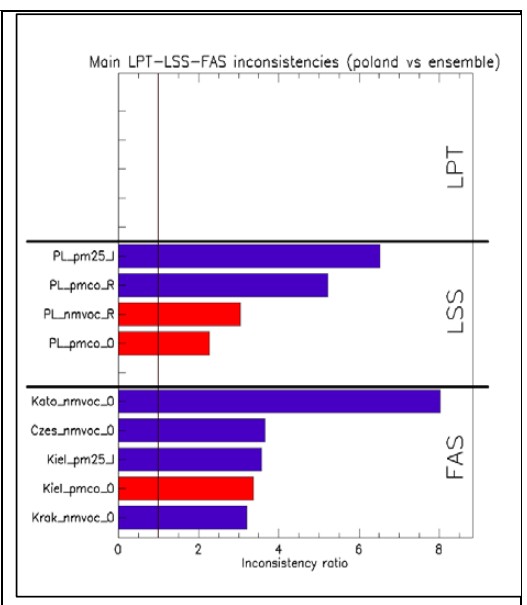

Figure 7: Diamond comparison of the local Polish vs ensemble inventory (left) and comparison of the ensemble top-down
components vs the ensemble restricted to the Polish territory.
At country scale, the largest inconsistency occurs for the industrial share for PM2.5 (factor 6
larger in the Polish inventory), for PMco and NMVOC from the residential sector (factor 5 lower
and factor 3 larger in the Polish inventory) as well as for PMco from the other sector (factor 3
lower in the Polish inventory). To support analyses on the country level, we present a
comparison of EMEP and CED country totals per pollutant for each GNRF sector analyzed as
well as some explanations for these differences (Table 2, supplementary material).
In the case of PM2.5, difference can be explained by the fact that the reports provided to NB are
based on user-specific permits which specify the list of pollutants to be reported whereas in EU
wide inventories, emissions are generally calculated using official EMEP/EEA emission factors.
In the case of NMVOC emissions, EMEP has higher values for all the sectors, with the exception
of residential combustion (GNFR C). The issue therefore originates from the sectorial share at
country level.
At the local scale (FAS), the spatial allocation of the NMVOC emissions for the other sector
leads to important differences in cities like Katowice, Czestochowa and Krakow. A similar



situation is found for PM in Kielce. We see from Figure 7 that this issue is general for all cities.
The large differences spotted in some cities (e.g. Kielce) for the "other" sector are caused by
emissions from heaps and excavations. While in CED, emissions from these sources are
accounted for, only emissions from brown coal excavations (part of NFR 1B1a) are included in
the EMEP inventory. These could explain the identified differences between the local scale and
Europe-wide ensemble inventory. Hence, including all heap and excavations emissions in EMEP
(and consequently in CAMS-REG) inventory would be advisable.
Relatively less important but yet about a factor 2 to 5, similar low values occur for the Power
plant $SO_2$ emissions (blue rectangles figure 7 left). None of the 3 Europe wide inventory shows
an inconsistency for these sectors/pollutants indicating a general issue between local and all EU-
wide inventories. The difference might be explained by the fact that CED is solely based on NB,
supplied directly with users' data, while Europe wide inventories (EMEP) likely include
additional emissions as they are based on overall fuel sales. In addition, point source emissions
from E-PRTR may be different from point source emissions used in national inventories, which
is also the case for Poland and may be therefore another source of inconsistency.
Another general issue is related to the PM residential emissions for which the Polish inventory
values are systematically larger than the ensemble ones for $PM_{2.5}$ and smaller for $PM_{co}$ which
can be partially explained by inclusion of condensable in CED. The EDGAR inventory differs
from the ensemble in a similar way and is therefore closer to the CED inventory values.
Although the magnitude of this inconsistency is less than previously mentioned ones, the size of
the symbols in the diamond diagram (Figure 7 left) indicate that the amount of PM2.5 emission
is important for that sector. The difference may be (partially) explained by the fact that the
EMEP and CAMS-REG inventories rely on versions of the official reported national inventories
from Poland that did not yet consider condensables as part of the PM2.5 emissions from small
combustion. In the 2022 submission, this was included and resulted in more than doubling of
total PM2.5 emissions from Poland as a whole. This will be included in future versions of
CAMS-REG and EMEP.. This is further addressed in the Discussion section.
The transport and industry sectors show the lowest number of inconsistencies (few points related
to those sectors in the diagram). While this is expected for transport which is a diffuse source,
this is surprising for the industry as this sector was the main source of inconsistencies at Europe
wide level. It is connected with the fact that the Polish EMEP reports, unlike CED, are based not
only on data provided by the users of NB.
The priority inconsistencies for each city are highlighted in Figure 8, and they are mostly related
to NMVOC for the "other" sector and PM for the residential sector. This is probably partially a
consequence of the processes behind the spatial allocation in the European-wide inventories.
While EU-wide inventory compilers distribute country totals obtained from bulk national
statistics, population density is often used as a spatial proxy. In this context, the resolution-free
design of CED inventory might be a paradoxical limitation here since the exact geographical
location of emission sources is prioritized, and some activities are very tough to allocate. For
example, coating applications (2D3d) which are responsible for >63 Mg of NMVOC emissions
(2018) in Poland, might be omitted in CED due to a lack of reliable spatial data in case they are
not provided by NB users in full. Yet another issue is that this pollutant is not being





monitored *in-situ* in Poland (and many other countries), which also hampers the interpretation of
emission data.

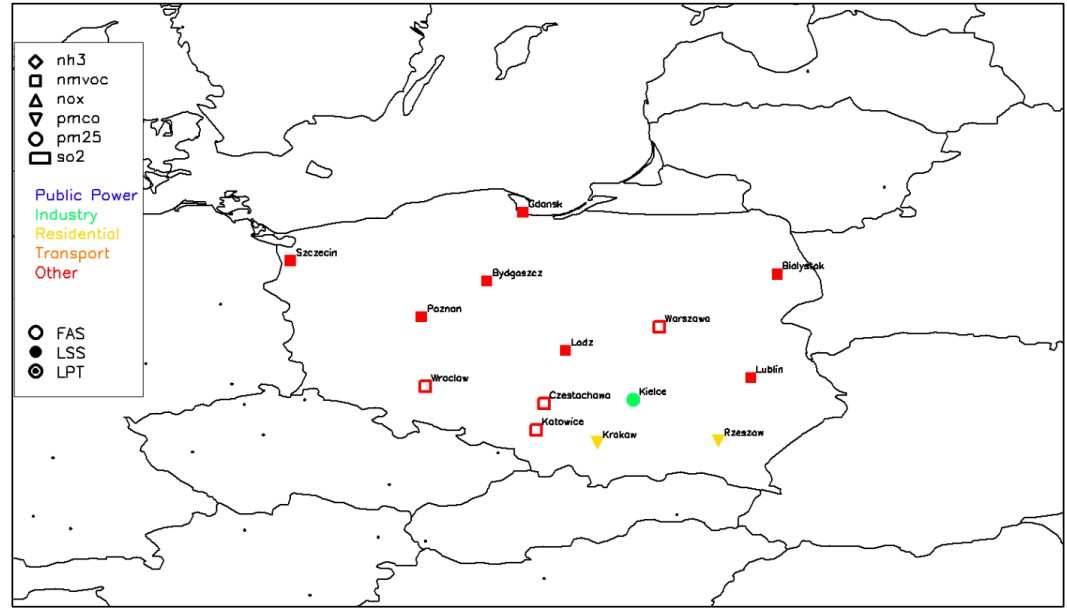

*Figure 8: overview of inconsistencies for the comparison between local emission inventory in Poland and the Europe wide*
*emission inventory ensemble*
In conclusion, the comparison of the Polish inventory with the ensemble mostly spots issues that
are related to a difference in terms of sectorial share at country level, explained by the
accounting of different sources in the two types of inventories. A similar argumentation can
explain part of the large discrepancies observed in some cities. Most of the issues occur for the
residential and "other" sectors and mostly for PM and NMVOC. Although the number of
inconsistencies may seem large, many of these are similar for all cities.

# 5. Discussion
European wide inventories are not totally independent of each other. Interlinkages between the
CAMS-REG, EDGAR and EMEP inventories have consequences for the comparison. For
example, EMEP is linked to CAMS-REG in that it (1) both inventories rely on country reported
data and (2) may use the same spatial proxies in case country do not report or the quality of the
reported data is poor. EMEP is also linked to EDGAR as it uses in some cases EDGAR
distribution as a proxy for gridding in case a Party is not reporting or the quality of the reported
data is poor (CEIP2022).
Part of the inconsistencies regarding Europe wide inventories are related to inconsistent values at
country scale. The comparison of EU-wide inventories highlights an important number of large
inconsistencies at country scale between EDGAR and the other two inventories: CAMS-REG



and EMEP. While the two latter use (but to different extents) officially reported emissions and
therefore rely on similar total emissions per country, EDGAR estimates emissions in an
independent manner, starting from activity levels and emissions factors from international
agencies and bodies (Crippa et al., 2018, Oreggioni et al. 2022). While this difference in
approach can explain a large number of inconsistencies identified for EDGAR, some of them are
very large, especially for $SO_2$ and PM in the industrial sector. For this particular sector, estimates
mostly come from the LPS and E-PRTR databases in EMEP/CAMS-REG, which emissions are
mostly based on measurements or facility-level estimates. Such information is not used in
EDGAR where estimates are based on fuel consumption and emission factors that are very
general and not plant specific. The screening analysis allowed identifying some of the causes
behind these differences (e.g. outdated sources and/or emission factors) that need to be improved
in EDGAR.
EU-wide, spatial inconsistencies mostly occur for the industry and "other" sectors.
Inconsistencies associated with EMEP and CAMS-REG mostly appear for the "other" and
industry sectors, mainly pointing to issues related to spatialisation, i.e. to urban activity shares.
The fact that the largest inconsistencies are found for sectors where point sources play a major
role was expected. Indeed, while a diffuse sector like transport may be distributed quite
differently, outliers would not appear as strongly as for point sources. A likely explanation for
the differences in $SO_2$ emissions is that their attribution to point sources is done only for those
included in point source reporting (E-PRTR). Smaller sources which are below the threshold for
E-PRTR reporting are distributed as diffuse sources to industrial zones (land cover class). This
may lead to over-allocation in some urban areas.
Local and EU-wide inventories are based on different emission estimation methodologies that
lead to inconsistencies in terms of sectorial share at country level. The reasons for
inconsistencies between local and European-wide inventories lays in different emission
estimation methodologies dictated by the primary purpose of these inventories. Based on
statistical data, commonly available in many countries, European-wide inventories rely on
general downscaled procedures to spatialize emissions, procedures that put a limit on the final
spatial resolution that can be reached for the inventory. On the contrary, local inventories like
CED are based on a bottom-up processes where the location and details of each source are
known. While we would therefore intuitively expect differences between local and European-
wide inventories to be driven mainly by spatialisation aspects, this is not always the case in our
analysis. Inconsistencies indeed relate mostly to differences in country sectorial shares that result
from different sectors/activities being accounted for in the two types of inventories. This is
particularly true for sectors like residential, industry or "others". As a result, for industry (GNFR
B), significant differences are noted for NMVOC, $PM_{10}$ and $PM_{2.5}$ (Table Supplementary
material 2).  For the residential sector, the main issue with European wide inventories is the use
of a generic approach for spatialisation over Europe, that neglects national and most important
subnational differences in the fuel energy mix. This is better captured in CED because of the
proxies that are based on local knowledge (see details in Gawuc et al., 2021).  In the case of
NMVOC in GNFR C, there are two possible reasons behind higher values in CED than the
ensemble. First, the larger share of coal in fuel mixes. Second, the higher values in emission
factors used in CED (see Table 2 in Gawuc et al. 2021).



Another reason likely to explain why spatialisation inconsistencies are minor is related to the fact
that EMEP reports for Poland, are gridded by Polish experts, utilizing spatial proxies based on
CED activity data for several sectors. This is the case in particular for stationary combustion
(GNFR C), road transport (GNFR F), and livestock (GNFR K). The last update was done in 2021
(Bebkiewicz et al.2022).
Many possible reasons for differences between local and Europe-wide inventories exist. In the
case of Poland, another possible source of inconsistencies between European-wide and local
Polish inventory is a consequence of how the Polish NB operates and under what rules. Any
given "user of the environment" is obliged to report emissions caused by a specific
industrial/chemical process for which his/hers "permit to use the environment" is issued. The
pollutants and GHG list that must be reported to NB differs among chemical/industrial processes
altering "users of environment" obligations. Emission from NB data is not taken into account by
the Polish National Statistical Office directly and the primary source of Europe wide inventories
activity data relies on national statistics. Furthermore, while the Polish EMEP reports are
partially based on NB and partially on original methodology (additional emission values) causing
disagreements with NB, CED directly adopts emission values reported to NB without additional
changes. This issue will be further investigated among CED and Polish EMEP compilers.
Yet another issue is that in the case of specific installations registered in NB, reports might be
based on direct stack measurements or actual condition of installations while the top-down
approach accounts only for general resources/fuel consumption. The advantage of NB over top-
down approaches is its sensitivity to temporal variability since reporting users are aware of any
changes in fuel or other resources quality they consume, rapid changes in production volumes,
new technologies used, newly mounted stack filters, etc. Those small changes might not be
captured in full in bulk national statistics, commonly based on fuel sales. Finally, it must be
commented that in the case of NB, the possible accidental "human factor" might be a source of
additional errors since reports are done manually via the online system. Despite some automatic
checking algorithms and manual expert evaluation, discrepancies are possible.
Finally it is interesting to note that the comparison between local and European-wide inventories
lead to additional inconsistencies than when the comparison is limited to Europe wide
inventories.
Uncertainties related to the screening methodology. As emission inventories are characterized by
different grid resolution and sector aggregations, some harmonization is required prior to the
screening process for a meaningful comparison. Conversion to a common grid resolution might
result in point sources shifted by one grid cell and be in the urban area in one inventory and not
in another, although having the same geographical coordinates in both inventory. However, the
city specific diamond diagrams can be used to check if this issue occurs.
## 6. Conclusions
The approach presented in this work is intended as a screening tool to flag inconsistencies among
inventories, and support the assessment of methods to estimate and spatially distribute emissions.



Only differences that are above a user-defined threshold are detected while smaller differences
are disregarded. This threshold reflects the limit between uncertainties and small errors on one
side, for which no emission inventory can be estimated to be the best because true emissions are
unknown, and larger differences on the other side for which we know that at least one inventory
has an error. Given the magnitude of the difference, in most cases this error is likely easy to
identify and that improvement in one or both inventories can be made, despite no real value is
known.
In this work, we created an ensemble inventory (median) with the aim of monitoring the status
and progress made with the development of Europe-wide inventories. Introducing an ensemble
also allows comparing many inventories at the same time in a relatively simple manner and
foster the interactions between emission inventory developers around the identified
inconsistencies. In contrast with other fields of applications (e.g. air quality forecast), this
emission ensemble is however not necessarily better than any of its components. While it is not
the more accurate inventory, it serves here as a common benchmark for the screening. In this
sense, the limited number of inventories (3) to create the ensemble is not a real issue in this work
although it should be kept in mind when analysing the details. The analysis of the ensemble and
the comparison with its individual components highlight a large number of inconsistencies.
While two of the three inventories behave more closely to each other (CAMS-REG and EMEP),
as to a large extent both inventories use emissions submitted to the CLRTAP as input data, they
yet show inconsistencies in terms of the spatial distribution of emissions. While the origin of
some differences between these inventories and EDGAR can be identified, their magnitude
remains to be explained. These differences mostly occur for $SO_2$, PM and NMVOC, for the
industrial and residential sectors, and reach a factor 10 in some instances. The screening results
provided useful information that allowed identifying necessary improvements on the estimation
of air pollutants emissions, in particular for EDGAR, with the PM emissions from the small-
scale combustion sector and $SO_2$ from the industry and power plant sectors.
The comparison with the local inventory for Poland leads to identifying another type of
inconsistencies. While one of the main differences between pan-European and local inventories
lies in the way emissions are spatialized, the identified inconsistencies do not relate to this
spatialisation process but are rather associated to the sectorial share at country level. These can
be also explained by the fact that there are different sources of data to calculate emission in local
inventory than in the European ones. In local inventory some emission sources are omitted due to
lack of the appropriate geographically allocated activity data, whereas are available on country
level e.g. industrial production. The screening identified some sectors and pollutants for which
discussion between local and EU-wide emission compilers would be needed in order to reduce
the magnitude of the observed differences (e.g. in the residential and industrial sectors)
The latter point is key. While it is more effective for inventory teams to meet and compare
approaches in detail to understand and correct differences between inventories, this can be
challenging at times, especially in the absence of a specific project to support the work. It must
however be noted, that in many instances the reporting of an inconsistency, especially when it is
very large, leads to a generally straightforward identification of the underlying cause without
requiring too detailed information regarding the inventories.



The settings used in this work, e.g. the choice of 150 urban areas or the way sectors are
aggregated are arbitrarily fixed. The methods allows for flexible choices and could be applied to
other areas than urban (e.g. high emission industrial or intensive agriculture areas) to assess the
consistency with respect to other types of emissions. In terms of sectors, a further disaggregation
of the "other" sector will be performed in future to better understand where inconsistencies
occur.
The ensemble is not meant to be a static entity. It will evolve as inconsistencies are progressively
discussed and solved. An ensemble is therefore associated with reference inventory versions as
well as with a reference year. The ECI and other statistics are provided to monitor progress and
point to potential improvements. In this sense the ensemble represents a useful tool to motivate
the community around a single common benchmark and monitor progress towards the
improvement of regional and locally developed emission inventories. It also ensures that
improvements become permanent, as forgotten improvements would indeed be flagged again by
the system.
While the comparison to one local inventory is presented in this work for example, these
comparisons can be systematized to improve the quality of the ensemble.
*Code and data availability.*
Supporting data and source code are available at: "Philippe Thunis. (2023). Supporting data for
the publication "Emission ensemble approach to improve the development of multi-scale
emission inventories" [Data set]. Zenodo. https://doi.org/10.5281/zenodo.7940402"
827 .
*Author contributions.*
PT and AC contributed to the study conception and design. Material preparation, data collection
and analysis were performed by PT, EP, ADM, JK, MB, LG, KS, and AC. All authors reviewed
the manuscript. All authors read and approved the final manuscript.
*Competing interests.* The authors declare that they have no conflict of interest.

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
