# Peer review of "Emission ensemble approach to improve"

_EGUsphere, 2023_

## Author Comment (AC1)

Thunis et al. present in their manuscript an ensemble (median) inventory to assess the quality of emission inventories. This new inventory is used to screen differences in terms of country totals, country sectorial share and share of the country emissions to the urban areas for emissions of several pollutants. For the screening, they use the method they already presented in Thunis et al. (2022).

There are clear differences between the current study and the former publication of Thunis et al. (2022). However, here no new technique is provided. What the authors actually provide is a new dataset that can be used to assess other emission inventories. Thus, I think that this manuscript does not fulfill the requirements for a publication in Geoscientific Model Development, but rather to a journal with focuses on data provision and/or application/evaluation of data sets.

In our view, the approach based on the ensemble inventory is more than a dataset and it modifies significantly the screening methodology. We agree, however, that the paper as written was not highlighting this sufficiently. We therefore substantially revised the manuscript to address this. In particular, we enlarged the section dealing with the construction of the ensemble and focused the analysis on the ensemble-based screening, highlighting the benefits of this methodology. We also moved to the supplementary material some of the bilateral comparisons to better stress the role of the ensemble. In addition, we restructured the discussion and conclusion sections with the former being now devoted to a discussion of the added value and limitations of the ensemble approach.

Further, the paper is not well written. Without reading the previous paper (Thunis et al., 2022) it is impossible to understand what has been done here; especially, which parameters are used for the assessment of the emission inventories and how to read the diamond plots.

We added explanations throughout the document (especially when introducing the methodology and visualization) to help interpreting the diamond diagram as well as the methodology.

The paper, especially, the results section, reads more like a scientific report than a scientific publication.

We substantially revised this section. The application of the approach is now centered on the ensemble methodology and the more technical paragraphs have been moved to the supplementary material (or removed) to focus on the key aspects.

Therefore, I would suggest to reject the current version and encourage the authors to revise and resubmit their manuscript to another journal.

**General comments:**

- The whole manuscript needs to be significantly revised. The abstract has not the structure an abstract for a publication should have and the result section reads more like a scientific report. The introduction does not give a clear overview over the topic. Further, all abbreviations need to be introduced and a list of abbreviations should be provided in the appendix since quite many are used here. I will provide more detailed comments under specific comments.

We significantly revised the manuscript and shortened it (about 200 lines less) to focus it on the ensemble approach. The abstract has been rewritten, the results section has been significantly shortened and re-structured putting emphasis on the added value of the ensemble (many paragraphs have been moved to supplementary material). We removed the more technical points and added a table of abbreviations in the appendix

- I think the usage of the term "ensemble" in your context is not correct. Here, it is rather confusing. Usually the term "ensembles" describes the performance of a model simulation several times with different initial conditions. However, what you are doing here, to my understanding, is averaging a certain number of emission inventories and providing a median value of these inventories that is screened (beforehand or afterwards?) to keep a certain quality standard. Thus, I would pick a different term than "ensemble".

We disagree with the reviewer. In many instances, ensembles are used for other applications than for multiple simulations under different initial conditions. In the field of meteorology, climate or air quality, different models are applied to the same set of initial conditions and the median defines the ensemble. The median is here defined for each sector and pollutant and for each city/country. This is the concept we use in this work. We tried to better explain it in the text.

**Specific comments:**

P1, L21ff: The method part is too complicated  and should be shortened. State more clearly what has been done in your study and what can your data be used for. I would suggest the following structure for your abstract: Start with why are emission inventories important, what are they used for and then describe what has been done in this study and what are the major results.

The abstract has been re-written to follow your suggestions.

P2, L53: The first sentence is hard to understand. Please rephrase. As stated in my general comments, what you are doing here is far away from what is done when ensemble simulations are used. Thus, this sentence rather confuses than helping to find an introduction to your study or the topic of emission inventories.

We simplified the sentence.

P2, L53ff: Start the introduction with how are emission inventories created, which ones are existing (give a short overview over the most important/known ones including references). What are the uncertainties of these inventories. What can be done or has be done to reduce the uncertainties and then describe your previous work and what has been done in this study.

We re-structured and rewrote part of the introduction to follow your suggestions.

P3, L113ff: Here, you describe how you compare the inventories, but shouldn't you first describe how you create your inventory?

We now included at start of the introduction a paragraph that provides background on how emission inventories are created and why they are uncertain. The starting paragraph reads as:

*Many studies have shown that emission inventories are one of the inputs with the most critical influences on the results of air quality modeling (Kryza et al., 2015, Zhang et al., 2015). Even more concerning, certain studies have shown that important uncertainties affect emission inventories, which may impeach conclusions based on air quality model results (Trombetti et al., 2018, Markakis et al., 2015). These uncertainties result from the need to compile a wide variety of information to develop an emission inventory. For the many pollutants and activity sectors to cover, the spatial and temporal distribution of emissions is typically based on proxies that can be estimated through different methods.*

P5, L200: How is the ensemble calculated. What actually do you understand under "ensemble"?

As mentioned in our response to the general comment, the ensemble is here understood as the median of three emission values. The median applies to each pollutant, sector, city and country. We clarified this in the paper.

P6, L221: Not clear, if you mean here the whole inventory or only a specific value.

It is for a specific sector-pollutant value. We clarified the sentence

P6, L232: Chose a more precise section title than just "Input data".

We changed it to "Required input emission data"

P6, 264-266: Sentence not clear. I think there is a doubling of what the difference between the two versions is.

We modified the sentence as follows: CAMS-REG-v5.1 is an update of version 4.2 that includes official national emission submissions for the year 2020.

P8, L308: Repeat here once again the GFNR sectors.

Done as suggested

P8, L310: Provide here an example or give more explanations what is meant with polygons.

We clarified the text as: "by aggregating gridded emissions on common polygons that delineate the area covered by a city or by a country. The city polygons used in this work are similar to those in Thunis et al. (2018)"

P8, L325: The abbreviation ECI has not been introduced and it should be more clearly stated what this parameter is used for or what it is describing.

The ECI indicator is introduced at line 171-172

P9, Figure 1: More explanation/guidance is needed to read the diamond plots. Further, in my opinion there is too much information in one figure.

We agree that there is many information in the diamond diagram. As we use this diagram for an overview, we prefer to keep it as is but we added explanations in the text to help interpreting the many parts of this diagram.

P9, L352-353: What is given by the numbers in parenthesis? The ECI? Also here more guidance is needed what the respective values mean.

No, the number within brackets is the total number of inconsistency associated to a given pollutant, sector or type. We added a sentence in the caption to clarify this. We also clarified it in the text.

P11, L412: It would be much more helpful if you could give the inconsistencies in percent. I still don't know how to judge these numbers. How many inconsistencies could be there?

The inconsistencies are given in absolute terms. Percentages can be obtained by dividing by the total number (given as NI). We clarified this in the text.

P13, L463: Since you do the application for Poland I would suggest to add Poland to the title.

Done as suggested

P13, L464: I would suggest to chose a more precise title than just "Input Data". Name the data sets used here, thus "The Central Emission Database".

Done as suggested

P13, L465-466: Why do you compare Poland with Europe? Wouldn't this be like comparing apples with pears? Are you selecting the Poland area from the European inventory to make it a correct comparison? Also here more explanations are needed.

We indeed compare the local Poland inventory with the EU-wide ensemble (extracting data over Poland). We clarified this in the text.

P14, L492: I already have forgotten what is denoted by A, B, D, E and J emissions. Which areas these letters are referring to could be repeated.

Done

P14, L503: Which cities and sectors have been chosen? Are these all located in Poland?

Yes, they are all in Poland. It is now clear from the modification of P13 (L465-466). We also made the link to Figure 8 for more clarity.

P14, L516: A better section header than just "Results" would be "Comparison of the CED inventory to the Ensemble".

Done as suggested

P17, L540-541: What do you mean here? Is that a repetition again? Check sentence and correct.

Corrected

P22, L753ff: Since you already have more than one page discussion, I do not understand why you need additionally more than one page of conclusions. The main

message and key points of your study are not coming through. These two sections need to be significantly improved.

We restructured these two sections. The discussion has been turned into a section focusing on the pros and cons of the ensemble approach. In both sections, we re-organised the existing paragraphs and moved some of them in the main text or in the supplementary material. We also paid attention to avoid redundancies.

P23, L767: Simple manner? You have not described at all how a user could download and use your data set. What requirements are needed, e.g disk space, software etc.?

What we mean here is that the preparation of the data is simple as is the screening methodology.

P23, L792:" .....whereas are available on country level". This sentence is not correct. Please check and rephrase.

Corrected

P23, L796: "The latter point is key" -> Please rephrase.

We removed that sentence

**Technical corrections:**

P6, L269: till nowadays -> up to date

Corrected as suggested

P7, L288: Abbreviation GFNR already on P6 introduced.

Removed

P8, L295: Abbreviation "EEA" introduced?

We included a list of acronyms

P8, L296: Same holds for "TNO", has this abbreviation introduced?

We included a list of acronyms

P8, L324: emissions ratios -> emission ratios

Corrected as suggested

P9, L347: No underlining of text is used in Copernicus journals.

Corrected as suggested

P10, L370: Has the abbreviation "LPT" been introduced?

Yes at line 134

P10, L376: Same for IEA.

We included a list of acronyms

P10, L378: of the -> for the

Corrected as suggested

P11, L398-399: Have the abbreviations "FAS" and "PMco" been introduced?

Yes for FAS (L132) and for PMco (L244)

P11, L405: appear -> appears

Corrected as suggested

P11, L408: Remove underlining of text.

Corrected as suggested

P11, L408: a factor 15 -> a factor of 15

Corrected as suggested

P12, L438: Remove underlining of text.

Corrected as suggested

P13, L450: what reported -> what is reported

Corrected as suggested

P13, L473: localization -> location?

Corrected as suggested

P14, L503: add "the" before "ensemble"

Corrected as suggested

P14, L503: I would rather use here "for" than "is" -> for 14 cities

Corrected as suggested

P17, L534: Sentence not correct. Please rephrase.

Rephrased

P19, L621: Add section number.

Corrected as suggested

P20; L655 and 663: Remove underlining of text.

Corrected as suggested

P21, L679 and 690: Same here.

Corrected as suggested

P23, L789: be also -> also be

Corrected as suggested

P23, L794: Full stop is missing.

Corrected as suggested

---

## Author Comment (AC2)

**Reviewer RC1**

The paper demonstrates two applications of using an existing emission inventory comparison method (Thunis et al. 2022) with ensemble concept introduced to illustrate inconsistencies in selected emission inventories. One application is for EU-wide emission inventory comparison and the other application is for local inventory emission inventory comparison. The paper is well-written and well-organized with detailed results and thorough analysis. The results are important and meaningful in terms of shedding lights on 'problematic' inventory with specific pollutant and sector combinations which would require attentions and check-ups from emission inventory developers.

However, the method used in the analysis lacks novelty even with ensemble concept introduced. The method has been described in detail in Thunis et al. 2022 and directly used in this paper with very limited modifications or improvements while the use of ensemble concept relates to input data not the method itself. Therefore, I would not recommend this work to be published on GMD. But I do believe the findings about emission inventory inconsistencies are important and provide insights on next generation emission inventory development. I would suggest the authors to re-organize the paper and submit to other journals which focuses more on applications and findings.

We substantially revised the paper to highlight the construction and application of the ensemble approach. We tried to stress the fact that the ensemble approach is more than just adding a new dataset. One main point is that using the ensemble reduces the number of bilateral comparisons, which relates to the methodology rather than to the dataset. We highlighted the novelty of the approach but also discussed its strengths and weaknesses. We made the following main revisions:

1) The application section has now been drastically reduced with several paragraphs removed or moved to supplementary material (SM). We focused in particular on the use of the ensemble approach, both at the local and European scales, and moved some of the bilateral comparisons to the SM. We also removed some of the most technical discussions.
2) We turned the discussion section into a discussion dedicated to the added value and limitations of the ensemble approach.
3) We introduced sentences throughout the text to clarify the added value of this approach, in relation to the introduction of the ensemble concepts

Comments on ensemble approach:

Though mentioned in discussion section, the limited number of ensemble members (3) might be problematic while typical ensemble approaches used in earth sciences in general requires more ensemble members, for example, WetCHARTS (Bloom et al. 2017b), a global wetland methane emissions dataset generated using bottom-up approach, has 18 ensemble members. Also, the approach of creating ensemble (taking median) with limited number of ensemble members, may result in selecting the same ensemble member for many [p, s] tuples so that the comparison is essentially inventory-to-inventory instead of desired inventory-to-ensemble.

We added the below paragraph to discuss this point in the "added value and limitations of the ensemble" section.

In our work, the number of members of the ensemble is limited to three. This would be an issue if the goal were to obtain more accurate and robust results with the ensemble. In such a case, the more members, the more robust the results of the ensemble. Our goal is however different and consists in creating a benchmark for comparison. Rather than looking at absolute values, we assess differences (between an inventory and the ensemble), for which the accuracy and robustness of the absolute values is of secondary importance.

Minor comments:

- The diamond diagram in the results section may cause confusion for readers, especially to those who is not familiar with Thunis et al. 2022.

We have added explanations in the text to help with the interpretation of this diagram

- Line 345: Is this line a separate figure caption?

No this is the introductory sentence of the paragraph. We removed it

- Figure 1 and following figures: what are the numbers in parentheses next to each legend items?

The numbers within brackets indicate the total number of inconsistencies for a given pollutant/sector. We clarified this in the caption and in the text.

---

## Author Response (AR2)

**Response to Reviewers**

Responses are in blue

All my comments have taken into account and the manuscript has improved significantly. I have only some minor technical issues that should be considered before publication:

P11, L405-409: This sentence is very long and somwhat complicated. I would suggest to split it into two sentences.

The sentence has been split and clarified as suggested

P12, L429: add "a" -> based on a

Done as suggested

P12, L447: I would rather write "led to identify" or just "let identify" dependent on what exactlly you want to say.

Done as suggested (led to identify)

P13, L487: Inclusion of what exactly? There is something missing after "condensable"?

We added "emissions" so that it now reads as condensable emissions

P13, 488-489: This sentence sounds also a bit incomplete. Please rephrase.

The sentence has been rephrased

P14, L519: add "the" -> " the supplementary material".

Done as suggested

P15, L550: Add "the" and change "do" to "does"-> when the country does

Done as suggested

P15, L551: with "Party" you mean the respective country? I would write here then country once again.

Done as suggested

P16, L568: inventory -> inventories

Done as suggested

P17, L598: add "of" -> a factor of 10

Done as suggested

References: Copernicus requires that in the reference list all authors are listed. Also instead of "&" "and" should be used. Please correct the reference list accordingly.

References have been reviewed, updated where relevant and harmonized to the GMD format